# The Trauma Recovery Actions Checklist: Applying Mixed Methods to a Holistic Gender-Based Violence Recovery Actions Measure

Laura Sinko [1], Limor Goldner [2] and Denise Marie Saint Arnault [3,*]

1   College of Public Health, Temple University, Philadelphia, PA 19140, USA; Laura.sinko@temple.edu
2   Emili Sagol Research Center, School of Creative Art Therapies, University of Haifa, 199 Abba Khoushy Ave., Mount Carmel, Haifa 3498838, Israel; limor.goldner@gmail.com
3   School of Nursing, University of Michigan, 400 North Ingalls Bldg, Ste. 2303, Ann Arbor, MI 48109, USA
*   Correspondence: starnaul@umich.edu; Tel.: +1-734-615-0537

**Abstract:** Gender-Based Violence (GBV) trauma recovery models have evolved in such a way that survivors are viewed as actively engaging in a multitude of strategies. In addition to seeking help and coping, survivors engage in diverse lifestyle, social, spiritual, and practical strategies to promote their health and wellbeing. This exploratory sequential mixed-methods study develops an instrument to measure the holistic recovery actions used by GBV survivors. The qualitative phase combined recovery action codes from interviews with 50 GBV survivors in three different survivor samples to create an initial six-concept 41-item Trauma Recovery Actions Checklist (TRAC). The quantitative psychometrics phase used data from 289 American GBV survivors. Results revealed a five-factor 35-item final version (sharing/connecting; building positive emotions; reflecting and creating healing spaces; establishing security; and planning the future). There were positive significant correlations between sharing/connecting and depression scores, and between sharing/connecting, reflecting, and building security with PTSD scores. No correlations were found between any recovery action type and the barriers to help-seeking subscales of Problem Management Beliefs, Discrimination, or Unavailability. However, there were significant negative correlations between Shame and Financial barriers and Sharing/Connecting, and between Feeling Frozen, Constraints, and Establishing Security. Implications for research, clinical practice and ways of understanding survivorship recovery are suggested.

**Keywords:** sexual assault; sexual abuse; rape; intimate partner violence; interpersonal trauma; trauma recovery; mental health; help-seeking; mixed-methods research





## 1. Introduction

Gender-based violence (GBV) is public health pandemic that involves the violation of human rights based on gender [1]. GBV is violence directed at individuals based on their biological sex or their gender identity and includes physical, sexual, verbal, emotional, and psychological abuse, threats, coercion, and economic or educational deprivation, whether in public or private life. According to the United Nations Population Fund, one in three women have experienced physical or sexualized violence in their lifetime [2]. When we also include the known emotional, financial, or verbal abuse, the numbers become even more staggering. While most research focuses on escaping violence or sexual violence prevention, research on the trauma recovery processes survivors engage in are emerging as a scientific research area. Theories and process models of recovery strategies [3,4] have evolved from viewing women as passive victims to viewing them as actively engaging in a multitude of private and public strategies to prevent, manage, and escape violence, and to recovery from trauma. If our prevalence estimates are correct, a third of women are recovering from GBV, and understanding their recovery actions is essential for research and intervention.

Trauma recovery actions cited in the literature primarily involve seeking to understand one's needs and communicating those needs with others to obtain help. In general, trauma recovery goes through three stages: problem definition and recognition, the decision to seek help, and then choosing the support resources one will use [4]. These support resources are generally categorized as informal (family, friends) and formal (the justice system, government agencies, religious leaders, shelters, and health professionals), and the assistance they offer can take various forms, including understanding, advice, information, and treatment [5]. Studies have shown that women who experienced GBV tended to prefer soliciting informal resources [6,7]. However, in persistent and escalating violence cases, women are more likely to use formal and more diverse services [8–10].

Cultural practices may also inhibit women's help-seeking. Some cultural practices that may impact help-seeking include degrading the survivor and their families' reputations, demeaning the women's social status, and social ostracism [11–13]. Feelings of shame, blame, fears of financial loss, and retaliation also prevent women from disclosing and seeking help from formal sources, and those who do seek help tend to seek help from family [7,14].

Extant literature tends to focus on informal support, service utilization and symptom management. Focusing only on those strategies may not capture the myriad of strategies survivors employ in their day-to-day lives to facilitate their recovery. This exploratory sequential mixed-method study used qualitative data to capture the recovery actions that survivors described, created a checklist to facilitate research that can capture these diverse actions, and conducted a preliminary evaluation of the resultant instrument. Rather than being driven by the service sector or by psychological theory, this method listens to the voices of the survivors to understand and capture what they value and do to recover, and then assesses the value of this instrument in a different sample of GBV survivors.

*The Trauma Recovery Actions Concept*

In addition to using formal and informal sources, survivors engage in trauma recovery actions that they hope can improve their relationships, increase meaning and life satisfaction, and improve their overall health. The concept of recovery actions may overlap with related concepts of help-seeking and trauma-related coping. Help-seeking has been defined as a process prompted by the perception of need or problem. It is characterized by problem-focused intentional action that involves interpersonal interaction with a third party [15]. However, others have argued that help-seeking is broader and may involve intrapersonal, lifestyle and spiritual actions. This help-seeking definition is any attempt to maximize wellness or to ameliorate, mitigate, or eliminate distress [16]. This definition is not trauma-specific but could encompass all actions taken by a survivor to decrease suffering, and to improve health and wellbeing, whether another person is involved or not. Our research focuses on trauma-related help-seeking, which is part of the broader domain of trauma related recovery actions.

The second concept that is similar trauma recovery actions is trauma-related coping. The coping literature is vast, and a full review is beyond the scope of this paper. Coping models are situated within the psychological literature and generally uses the Lazarus and Folkman Cognitive process of coping model [17,18], which centers on how coping responses are framed by an individual's cognitive appraisal of stressors. Here, we confine ourselves with trauma-related coping, which situates trauma as the psychological stressor. One notable study in this regard is a recent meta-analysis of 39 studies that evaluated the relationships between the use of approach and avoidance strategies (both problem-focused and emotion/cognitive focused) following interpersonal violence or severe injury trauma and psychological distress outcomes. Problem-focused or emotion-focused coping strategies can be further subdivided into coping strategies that directly address the stressor and those that are aimed at avoiding the stressor. Overall, this metasynthesis found a consistent association between avoidance coping and distress ($r = 0.37$), but no association between

approach coping and distress [19]. A related line of research investigates the impact of trauma-related coping self-efficacy on PTSD outcomes finds similar results [20,21].

The coping literature uses adaptive and maladaptive coping concepts, aiming to predict outcomes of coping types and make treatment recommendations. Unfortunately, however, the use of these concepts may have the unintended consequence of pathologizing trauma recovery in a few ways [22]. GBV impacts survivors at multiple levels (mental, emotional, spiritual, and social). GBV often involves cumulative impacts on all those levels. Moreover, many forms of GBV are not single events but are often part of a field of events or a dynamic pattern exerted across time. Therefore, focusing on psychological responses and mental health outcomes is incomplete and neglects the complex interactions among the breadth of interacting variables that shape recovery responses. Second, from a philosophical perspective, the impact of trauma on a person should not be construed as their psychological problem, but rather as a natural and expected response to betrayed trust, painful violations, and broken relationships. Third, when psychology labels recovery processes as some kind of "impairment" or "maladaptive" response, they place the problem at the feet of the survivor, reducing them to the "thing that happened to them". We conclude that exclusive problem-oriented psychological approach can pathologize the survivor by framing their suffering as caused by their inappropriate or problematic responses to their violation, rather than seeing trauma recovery actions as a dynamic and multidimensional range of attempts of a holistic being who is continuously trying to better themselves and heal after emotionally devastating violations. Our research is, of course, looking at help-seeking and coping as part of the broad array of cognitive, emotional, social, and spiritual responses and actions to promote recovery, but we begin with the survivor's perspective rather than any specific psychological frame.

Beyond help-seeking and coping. In addition to seeking help from informal and formal sources, or approach and avoidant coping, our research has found that GBV survivors engage in many complex and diverse behaviors to promote their health and wellbeing [23–27]. In addition, other quantitative and qualitative studies have demonstrated the contribution of spirituality to decreased PTSD and increased resilience [28–30]; advocacy, altruism, and helping other women [31,32]; engaging in calming, nurturing sports and healthy activities [33,34]; reconnecting with family, peers, and community members [28,31]; navigating new romantic relationships and exploring femininity [34,35]; engaging in self-exploration; rediscovering interests, desires, and skills [31,36]; engaging in creative activities [31,33]; and creating stability through employment, housing, and parenting [29,36].

Taken together, this review points to the need for a holistic survivor-centered trauma recovery actions definition, as well as an instrument to capture these actions. Despite this robust literature, there are no empirical instruments to date that attempt to quantify the number and type of trauma recovery actions survivors engage with on their healing journey. This study used an exploratory sequential mixed-methods design to define and develop a measure to capture diverse trauma recovery actions and evaluate the relationships among trauma recovery actions, help-seeking barriers, and symptom burden for survivors of GBV.

## 2. Materials and Methods

This research represents an exploratory sequential mixed-methods study aiming to develop a survivor-based instrument to measure recovery actions in survivors of GBV. To carry out the qualitative phase of this study, we used qualitative data from our larger international study that focused on recovery for trauma survivors called the Multicultural Study of Trauma Recovery (MiStory). We examined the recovery action codes and quotations from previously analyzed English language qualitative interviews from 50 GBV survivors in different survivor samples to create the Trauma Recovery Actions Checklist (TRAC). The quantitative phase psychometrically evaluated the TRAC to identify factors, and reduce items, and established predictive validity using symptom burden and help-seeking barriers scales ($N = 289$).

*2.1. Sample*

Qualitative phase. We recognize that recovery actions will vary depending on age, social context, cultural context, type of violence, personality factors, religious affiliation, access to recovery support services and a host of other variables. Therefore, we chose to use qualitative datasets from a variety of GBV survivors. This strategy gives us the ability to hear about the recovery actions of sexual assault survivors, intimate partner survivors, and people who had complex trauma. To do this, a dataset was created by merging recovery action codes from the qualitative datasets from three studies that used the same research and analytic design to understand the GBV recover in different survivor samples (*N* = 50). The qualitative dataset across the three studies consisted of interviews following the Clinical Ethnographic Narrative Interview CENI protocol [37]. All research was approved by the University of Michigan IRB (HUM00144780 for the sexual assault survivors; HUM00091662 for all others). All respondents self-identified as women and as GBV survivors. One sample was 19 sexual assault US survivors ages 18–26 [27,38]. Twelve of these women were current undergraduate students at a four-year institution, and seven women were alumni. Thirteen identified as Caucasian, three women identified as African American or Black, and three identified as Asian. Eight women had a history of childhood abuse, 16 women had a history of unwanted sexual intercourse when they were enrolled as undergraduates, and 18 women disclosed other forms of unwanted sexual contact as undergraduates. Another sample included 19 GBV survivors from Southeastern Michigan, ranging from 20 to 81 years, and about one-half had children [39]. Eighteen of the women identified as Caucasian, two women identified as African American, and one woman identified as Asian. Half of the sample had experienced more than one type of GBV; five had experienced child abuse, ten had experienced sexual abuse, and nine had experienced domestic abuse. The third sample included twelve Irish domestic abuse survivors [25]. The Irish women were primarily from a rural area of Ireland, ranging from 20–64 years in age, and all had children. Eleven women identified as Caucasian, and one identified as Black. All but one of the women was born in Ireland; however, several had lived abroad, primarily in England. All twelve participants reported instances of abuse (emotional, physical, or sexual) in childhood and domestic abuse. Seven of the nine participants who identified abuse in childhood stated that their abuser was their father.

Quantitative Phase. The evaluation of the TRAC was carried out using data from 276 GBV survivors who responded to an anonymous online survey advertised through a clinical health research portal (HUM00191183). All respondents self-identified as women and as GBV survivors. The ages ranged from 18–61 (M = 38, SD = 14.6). Forty-six percent of these survivors had children. The types of violence in the survey were not mutually exclusive, so the types of violence are physical violence (55%), sexual violence (65%), psychological violence (80%), stalking (36%), and economic abuse (17%). Eighteen percent endorsed that they experienced GBV within the last year but that it was over at the time of the survey, and, sadly, six percent were currently experiencing GBV. About half of the sample were employed full-time (46%) or part-time (16%). Eighteen percent were either retired or permanently disabled, 14 percent were students, and nine percent worked in the home in caregiving and parenting. Ninety percent had some college or a degree. Six percent were immigrants. We did not ask about race, ethnicity, preferred language, or specific gender identity.

*2.2. Instruments and Measures*

**Qualitative instruments.** The Clinical Ethnographic Narrative Interview (CENI) was used to collect data for the study [37,40,41]. The CENI is a semi-structured interview adapted to study trauma recovery. The CENI lasts about 90 min and utilizes four unique participant activities (social network, body map, lifeline, and card sort) to investigate social and cultural experiences, beliefs, barriers, and facilitators to healing from GBV. The interview begins with a social network map to frame help-seeking within the social context. Next, we invite the participant to use body mapping to place their distress onto their

body and focus on their internal process. Then, the participant completes a retrospective overview of triumphs and distress in their life in a lifeline to find patterns and link past and subsequent events, emotions, and recovery actions. Finally, the participant completes a card sort referencing her most recent low point to describe her distress and healing in detail on a focused event, creating a map of her symptom clusters. These cards are examined in terms of the survivor's interpretations of causes, consequences, meaning of healing and help-seeking actions taken. Throughout the interview, information about help-seeking and recovery actions taken by the survivor are discussed, and this information was used to develop the recovery actions tool described in this paper.

**Quantitative Instruments.** Symptom burden was measured with the PHQ8 depression scale and the PC-PTSD screener. The PHQ8 is an eight-item depression scale that does not include the suicidal thoughts item in the PHQ-9, making it suitable for survey use [42–44]. A score of 10 or higher on the PHQ8 indicates probable major depression. Cronbach's reliability in our study was 0.88. We also used the Primary Care-PTSD (PC-PTSD) screen that includes four questions about the presence of PTSD symptoms [45]. Cronbach's alpha reliability for the scale in this sample was 0.73. There are barriers to help-seeking after GBV (BHS-TR). Our 34-item scale used the original 25 items in the Ontario Barriers to Care scale [46,47] plus an additional nine trauma-specific items developed by GBV survivors [24,48]. The BHS-TR asks about barriers to seeking help for GBV recovery in the past year. Our instrument's directions are "Think about your experiences and feelings that are a result of Gender-Based Violence. In the last year, how much of the feelings or attitudes listed below influenced your decisions not to seek help?" The responses were on a 4-point Likert scale anchored at 1 (Did not influence me) to 4 (Strongly influenced me). The BHS-TR includes internal barrier subscales including shame, feeling frozen, and problem management beliefs. It also includes structural barrier subscales including finances, perceived discrimination, perceived availability, and external constraints. Cronbach's alpha reliability for the total scale in this sample was 0.89.

### 2.3. Analysis

Phase 1 was a qualitative analysis of the codes and quotes about recovery actions after GBV from survivors in previous studies led by the first and senior authors. Grounded theory analysis was used to identify recovery actions and overarching characterizations [49,50]. Recovery actions in the qualitative analysis were defined broadly as any action survivors carried out to help manage their symptoms or to move towards healing and recovery. Close readings of the transcripts contributed to a preliminary sense of the interviews, and a preliminary code list of the discovered recovery actions was developed. Central themes were identified through a process of abstraction, in which the analyst "lumps" similar substantive or grounded phrases given by the participant codes, and then gives them a more abstract conceptual name or thematic descriptors. This process allows for "systematic comparison" and "conceptualizing". The codes under each categorization were created into initial checklist items using participant wording ($n = 67$). These were grouped into the initial themes list that served as the initial recovery action categorizations ($n = 6$). Next, because there were 67 items under six themes, we tried to reduce the number of recovery actions by synthesizing and condensing items. This was conducted in an iterative fashion, with the lead and senior author meeting to reconcile differences and confirm shared understanding. This process resulted in a final measure of 41 items under the six initial conceptual themes. ATLAS.ti qualitative software was used for data management and analysis [51]. An audit trail using personal, theoretical, and analytic memos was maintained, with coding concepts discussed at length in research team meetings to verify accuracy.

Phase 2 was a quantitative analysis of the recovery actions. Further data reduction and reliability was examined using exploratory factor analysis. Descriptive analysis of the sample was carried out. Predictive validity was then carried out by examining the relationships between the TRAC subscale scores, depression and PTSD symptoms, and

barriers to help-seeking. We predicted that higher distress and more perceived barriers to help-seeking would be associated with fewer trauma recovery actions.

## 3. Results

### 3.1. Qualitative Phase: Instrument Development

Our qualitative research discovered six main categories of trauma recovery actions that GBV survivors engaged in as they navigated their healing process. These were lifestyle self-care, emotional self-care, relating to others, finding safety, finding peace, and futuring.

Lifestyle self-care. Participants described many actions they used in their daily lives that promoted their physical and mental balance. These included making lifestyle changes, finding rest and relaxation, creating and sticking to a routine, taking breaks to heal, and scheduling times for things that bring balance into life. For example, one survivor shared the importance of relaxation, saying, "you know [it's important]. Taking the time to put all the stress and the to-do list on the back burner and just enjoy relaxing for a moment, as well as remembering that it's okay to relax" (Participant AK0112). This quotation, along with other quotations relating to relaxation, led to creating the item "Making sure I relax and rest my mind and body regularly". Five items were created for this group of recovery actions.

Emotional self-care. In addition to caring for one's physical health, caring for one's personal and emotional needs were also important. This self-care included looking inwards to challenge negative thinking and understand one's boundaries, engaging in activities that release emotions or help reconcile the past, and performing actions that nurture one's emotional and spiritual self. For example, one person reflected on the importance of setting boundaries in her life and how she wished it were taught more at a young age, saying: "Is there a way at a school-age level that we could talk about healthy boundaries? We talk about stranger danger; why can't we talk about healthy boundaries in general? Why can't we make it bigger and help young people because they may be in families that are bankrupt on emotional stability" (Participant A079). This quote, along with other quotations about boundary setting, led to creating items such as "Communicating my needs and boundaries". Nine items were created for this group of recovery actions.

Relating to others. Spending time with others was another type of recovery action that survivors mentioned. Outside of spending time with friends and other loved ones, relating to others consisted of learning how to ask for help, build new relationships, sharing their GBV experience, and care for others. For example, one participant discussed relating to others in the context of caring for her children, saying, "My children, there is the reason for me living. So, when I wake up in the morning and feel whatever, I look at my kids and see that they are happy and that makes me happy" (Participant I012). This quotation, along with other quotations that discussed caregiving, led to the creation of items such as "Caring for those whom I love (i.e., child, pet, family member)". Seven items were created for this group of recovery actions.

Finding safety and justice. Survivors described recovery actions which included engaging with professional services, securing a safe and nurturing home, engaging in education, and taking actions to improve one's feelings of security. For example, a participant shared, "I bought my own home which my daughter said to me, 'mam if you have to rent for the rest of your life if you have to be it so,' but it's still a sense of security to have your own" (Participant I012). This type of comment led to creating items such as "Securing or creating a safe and nurturing home". Five items were created for this group of recovery actions.

Finding peace, joy, and contentment. Another thing the survivors sought to cultivate was a sense of peace. These recovery actions included engaging in faith, seeing the beauty in nature, practicing mindfulness and gratitude, and finding playful or fun times. For example, one participant shared the importance of experiencing joy despite what she has been through, saying, "taking the time to be goofy and ridiculous and not be serious all the time and not worry and not necessarily ignore problems but, um, not let them take over and be overwhelming [is important] . . . just allowing myself to be happy and enjoy

the silly moments" (Participant AK0112). This quotation contributed to the creation of the items such as "finding times to be playful/silly/ laugh/have fun". Five items were created for this group of recovery actions.

Futuring. Finally, participants ultimately sought to engage in actions that helped them build a new future, separate from their violence experiences. These recovery actions included trying new things, setting short and long-term goals, applying for financial support, and furthering their development with training or additional education. For example, one person shared, "I would say like my definition of healing is, um, I think planning. A lot of it is planning for your future" (Participant AN2414). This participant continued, "[it's helpful to have] goals. Short-term goals . . . long-term goals . . . And just paying attention to myself because I often put other people above me". This set of comments contributed to creating items such as "creating a vision for the future (e.g., long-term goals)". Six items were created for this group of recovery actions.

Based on the results of the qualitative phase of this study, we define Trauma Recovery Actions as: the holistic array of personal, social, and spiritual actions used to create positive emotions, facilitate positive relations, and foster meaning, purpose, peace, and hope.

### 3.2. Quantitative Psychometrics Phase

Sample Characteristics. A total of 276 GBV survivors took part in phase 2 of this study (see Table 1). The average age was 33 years, and the majority were employed, with over half having received some college education, and 68% having a college degree. Racial or ethnic data were not requested; however, sixteen women indicated they were born in a country other than the U.S. About half of the sample had children. Sixty-seven had experienced GBV within the last year, and over half reported physical or sexual violence, and over 80% reported psychological violence. Fifty-four percent of the sample were above the clinical threshold for depression on the PHQ8 scale, and 42.8% were above the threshold for the PTSD screener.

**Table 1.** Sample characteristics.

| Demographic Variable | M (S.D.) |
|---|---|
| Age (Missing = 29) | 33 (14.7) |
| PHQ8 | 10.3 (5.5) |
| PTDS Screen | 2.1 (1.5) |
| Employment (not mutually exclusive) | |
| Working | 172 |
| Unemployed, looking for work, Working in the home | 47 |
| Student | 41 |
| Retired or Disabled | 53 |
| Education | |
| Some college | 21.7% |
| College grad | 38% |
| Grad courses or Grad degree | 30% |
| Children | 45.7% |
| Reported violence type (not mutually exclusive) | |
| Physical Violence | 55.1% |
| Sexual Violence | 64.9% |
| Psychological Violence | 81.5% |
| GBV in the last year | 231 |
| Yes | 16 |
| Yes, but now it's over | 51 |
| No | 202 |

Factor Analysis. Participants were given the resultant 41-item checklist and were asked to check all the actions they were currently using (See Table 2 for final measure). Because we had dichotomous data, we carried out exploratory factor analysis (EFA) in MPlus [52] using the GEOMIN rotation [53,54]. The GEOMIN rotation is recommended for producing

factor loadings and factor correlations without indicating the factor-loading pattern [55]. One to five factors were extracted. Model fit statistics provided in MPlus were used to compare fit for the models. The Chi-Square tests the possibility that the model does not fit significantly worse than a model where the variables correlate freely; *p*-values greater than 0.05 indicate a good fit. The Root Mean Square Error of Approximation (RMSEA) adjusts for model complexity, making it sensitive to the number of parameters in the model [54] with values of less than 0.06 reflect a good fit [52,56]. The SRMR assesses the average residuals for the correlation matrix and recommends less than 0.05 [53]. The Tucker–Lewis Index (TLI) [57] and the Comparative Fit Index (CFI) [58] compare the user-specified model to a baseline model, with values greater than 0.95 indicating a good fit [56,59]. Items with a factor loading lower than 0.30 were excluded from further analyses [60]. The analysis found an advantage to the five-factor solution (for the model's comparison, see Table 2).

**Table 2.** Factor Loadings for the Trauma Recovery Action Checklist (TRAC).

| Cronbach's alpha reliability for entire checklist = 0.89<br>Chi-Square Test of Model Fit Value 433.410, \Degrees of Freedom 430, *p*-Value 0.130 | F1 | F2 | F3 | F4 | F5 |
|---|---|---|---|---|---|
| **F1: Sharing and Connecting** (Eigenvalue 12.830, 9 items, Cronbach's alpha = 0.67) | | | | | |
| 6: Sharing my gender-based violence experiences in healing spaces | 0.61 | | | | |
| 8: Communicating my needs and boundaries | 0.60 | | | | |
| 9: Recognizing and challenging harmful thoughts | 0.49 | | | | |
| 15: Asking for help and letting others help me | 0.44 | | | | |
| 20: Finding ways to play as a way to connect | 0.41 | 0.36 | | | |
| 34: Finding times to be playful/silly/laugh/have fun | 0.37 | | | | |
| 23: Sharing my story publicly | 0.74 | | | | |
| 24: Using trauma-related services/education | 0.50 | | | | |
| 26: Learning about/engaging to promote human rights | 0.52 | | | | |
| **F2: Building Positive Emotions** (Eigenvalue 3.582, 4 items, Cronbach's alpha = 0.52) | | | | | |
| 5: Scheduling time for enjoyment | | 0.45 | | | |
| 18: Connecting with loved ones and family | | 0.50 | 0.35 | | |
| 7: Engaging in activities that release strong emotions or feelings | | 0.31 | | | |
| 10: Expressing myself through writing or creative arts | | 0.53 | | | |
| **F3: Reflecting and Creating Healing Spaces** (Eigenvalue 1.878, 12 items, Cronbach's alpha = 0.78) | | | | | |
| 4: Spending time alone | | | 0.49 | | |
| 2: Resting and relaxing my mind and body | | | 0.60 | | |
| 11: Finding or creating healing spaces where I can recharge | | | 0.41 | | |
| 12: Bringing beauty, strength, and pride back into my life | | | 0.50 | | |
| 13: Taking needed space and breaks to heal | | | 0.51 | | |
| 14: Avoiding activities or spaces that are harmful to me | 0.35 | | 0.36 | | |
| 31: Practicing religion; engaging in my faith; praying; connecting with a higher power | | | 0.32 | | |
| 16: Building and engaging with friends | | | 0.46 | | |
| 32: Being in nature/finding beauty in nature; gardening | | | 0.54 | | |
| 33: Practicing being mindful/fully present; Breathing deeply/consciously | | | 0.64 | | |
| 39: Organizing your space | | | 0.55 | | |
| 35: Practicing gratitude/acceptance/self-acceptance | | | 0.90 | | |
| **F4: Establishing Security** (Eigenvalue 1.792, 4 items, Cronbach's alpha = 0.65) | | | | | |
| 27: Reporting to police/authoritative body | | | | 0.82 | |
| 28: Preparing for legal action/gathering evidence | | | | 0.81 | |
| 29: Taking legal action to protect yourself, children, or property | | | | 1.02 | |
| 30: Taking actions to improve my feelings of safety and security | | | | 0.56 | |
| **F5: Planning the Future** (Eigenvalue 1.472, 6 items, Cronbach's alpha = 0.70) | | | | | |
| 19: Exploring new romantic relationships | | | | | 0.48 |
| 36: Trying something new that is inspiring or challenging | | | | | 0.48 |
| 37: Creating a vision for the future (e.g., long-term goals) | | | | | 0.79 |
| 38: Preparing with training or education | | | | | 0.68 |
| 40: Applying for financial support/jobs | | | | 0.40 | 0.55 |
| 41: Setting short term goals; taking small steps; celebrating small victories | | | | | 0.50 |

In total, six of the original items loaded weakly (≤0.30) in the analyses (items #1, 3, 17, 21, 22, and 25), so these items were removed from subsequent EFA iterations. We set our cross-loading threshold above 0.35. Item 20 had a more robust primary loading of 0.41 and

a clear conceptual link, so it was retained for the final model. The final model reduced our initial six conceptual themes into five components based on 35 of the original 41 items. The factors that emerged were: Sharing/Connecting (nine items), Building positive emotions (four items), Reflecting and creating healing spaces (12 items), Establishing security (six items), and Futuring (six items). The Cronbach's alpha measure of internal consistency for the 35-item scale was 0.89. Cronbach's alpha of the sub-scales and the complete 35-item scale are shown in Table 3, descriptive results for the subscales are in Table 4, and the correlations between the factors are shown in Table 5.

**Table 3.** Model Fit Indices Comparison.

| Model | $\chi^2$ | *df* | *p* | RMSEA | CFI | SRMR | TLI |
|---|---|---|---|---|---|---|---|
| 1 factor | 1198.064 | 779 | <0.0001 | 0.043 | 0.893 | 0.139 | 0.888 |
| 2 factors | 908.460 | 739 | <0.0001 | 0.028 | 0.957 | 0.103 | 0.952 |
| 3 factors | 830.231 | 700 | <0.0001 | 0.025 | 0.967 | 0.091 | 0.661 |
| 4 factors | 758.158 | 662 | <0.001 | 0.022 | 0.976 | 0.085 | 0.970 |
| 5 factors | 691.472 | 625 | 0.033 | 0.019 | 0.983 | 0.073 | 0.978 |
| 5 factors * | 433.410 | 430 | 0.13 | 0.016 | 0.991 | 0.066 | 0.989 |

* Excluding 6 items 1, 3, 17, 21, 22, and 25.

**Table 4.** Factor descriptive data.

| Factors | *M* | *SD* | Min. | Max. |
|---|---|---|---|---|
| F1: Sharing/Connecting | 2.89 | 2.14 | 0.00 | 9.00 |
| F2: Building Positive Emotions | 1.65 | 1.23 | 0.00 | 4.00 |
| F3: Reflecting and creating healing spaces | 6.11 | 3.12 | 0.00 | 12.00 |
| F4: Establishing Security | 0.54 | 0.90 | 0.00 | 4.00 |
| F5: Planning for the Future | 1.97 | 1.72 | 0.00 | 6.00 |
| *Total* | 13.17 | 7.11 | 0.00 | 35.00 |

Note: *N* = 276.

**Table 5.** Correlation matrix.

| | Sharing | Building Positive Emotions | Reflecting and Building Healing Spaces | Establishing Security | Futuring |
|---|---|---|---|---|---|
| Sharing | | | 0.62 ** | | 0.52 ** |
| Positive Emotions | 0.48 ** | | | | 0.43 ** |
| Reflecting | 0.62 ** | 0.57 ** | | | 0.56 ** |
| Security | 0.30 ** | | 0.29 ** | | |
| Futuring | 0.52 ** | 0.43 ** | 0.56 ** | 0.22 ** | |
| Barriers to help-seeking subscales | | | | | |
| Shame | | −0.16 ** | | | |
| Financial | | −0.14 * | | | |
| Frozen | | | | 0.15 * | |
| Constraints | | | | 0.16 * | |
| PHQ Sum | 0.13 * | | | | |
| PC-PTSD Sum | 0.26 ** | | 0.15 * | 0.19 * | |

Note. *N* = 227, ** $p \leq 0.01$ * $p \leq 0.05$.

Predictive validity. Table 5 presents the intercorrelations between the factors of the recovery actions demonstrating moderate significant positive correlations. The associations between recovery actions and perceived barriers to help-seeking were less straightforward. No correlations were found between any trauma recovery action type and Problem management beliefs, Perceived Discrimination, or Perceived Availability. However, there were weak significant negative correlations between the TRAC subscale Sharing/Connecting and the BHS-TR subscales of Shame and Financial barriers. There were also weak positive significant correlations between the TRAC subscale Establishing Security and BHS-TR subscales of Feeling Frozen and Constraints.

A few associations between recovery action types and symptom burden were revealed. There were small positive significant correlations between sharing/connecting and depres-

sion. There were also small positive significant correlations between sharing/connecting, reflecting, and building security with PTSD screening sums. *T*-tests analyses revealed no differences in the mean recovery action sums between survivors who met the criteria for depression ($n$ = 150) and those that did not ($n$ = 126). However, there were differences in the means for those who screened positive for PTSD ($n$ = 118) compared with those who did not ($n$ = 158). Specifically, women with suspected PTSD had higher mean scores for sharing/connecting (M = 3.46, SD = 2.21) compared with those who did not screen positive (M = 2.50, SD = 2.0) t = −3.9 (df = 274), $p$ = 0.00. Women with suspected PTSD had higher mean scores for reflecting (M =6.58, SD = 3.16) compared with those who did not screen positive (M = 5.76, SD = 3.11) t = −2.1 (df = 274), $p$ = 0.03. In addition, women with suspected PTSD had higher mean scores for establishing security (M = 0.73, SD = 1.0) than those who did not screen positive (M = 0.39, SD = 0.76) t = −3.1 (df = 274), $p$ = 0.02. Finally, women with suspected PTSD had higher mean scores for futuring (M = 2.11, SD = 1.76) compared with those who did not screen positive (M =1.80, SD =1.68) t = −2.0 (df = 274), $p$ = 0.05.

## 4. Discussion

This study describes the development and evaluation of the recovery actions checklist (TRAC) to measure a holistic range of recovery actions described by GBV survivors. This measure responds to the dozens of anecdotal comments from GBV survivors stating that they have maintained their health and security in numerous ways outside of informal self-disclosure and formal service use. Of course, many survivors engage with the health and mental health system. However, we have heard repeatedly in interviews that survivors feel these systems often focus solely on their symptoms. Moreover, this scale captures the variety of ways that survivors go about recovery, representing their resilience, survivorship, and engagement in recovery actions on a day-to-day basis. Much of the literature on recovery interventions from GBV has had a crisis-oriented perspective [61,62]. Interventions are often aimed at increasing women's safety [33,34], improving women's functioning by lessening their trauma symptoms [63], or increasing women's awareness of their victimization and "devictimize" them by encouraging them to leave the abusing partner [64,65]. Nevertheless, a body of literature shows the range of recovery actions survivors are engaged in, affirming that the GBV recovery process is multifaceted and includes broader goals than attaining security. In fact, our findings indicate that women reconcile their abuse history by navigate complex emotional, bodily, and social processes, engaging in future orientation, and establishing emotional security and safe spaces. These findings are supported by a recent qualitative metasynthesis of survivor healing goals which includes key objectives of (1) trauma processing and re-examination, (2) managing negative states, (3) rebuilding the self, (4) connecting with others, and (5) regaining hope and power [26].

The relationships between recovery actions and symptom burden were interesting. The positive relationships between recovery actions and PTSD screener scores suggest that survivors with PTSD symptoms are engaged in recovery actions, taking charge in their lives and being agentic in their recovery [33]. This recovery agency view is a different picture from the endless articles that focus only on the symptom burden for survivors and is in concert with the newer, positive, and empowering literature about survivorship and thrivership [66,67]. In contrast to surviving, which means "to continue to live or exist", and "resilience", which refers to the ability to cope with uncertainty by bouncing back to the initial condition while avoiding adverse outcomes, our finding suggests that women thrive to higher-level functioning that reaches beyond gaining homeostasis following the adverse event [68,69]. Moreover, posttraumatic growth is characterized by positive future orientation. While thriving, posttraumatic growth and recovery actions are distinct concepts, they overlap somewhat. Regardless, our findings concur with a definition of thriving and posttraumatic growth as characterized by a positive future perspective, improved health, and wellbeing, and reclaiming of the self [66,70].

Many of these recovery actions might be termed "passive" coping, but we believe that a pejorative label suggests that these internal actions are less necessary or not agentic. From an empowerment perspective, practitioners need to value and promote and enable a range of coping responses. Moreover, meaning-centered actions are essential in restoring self-love and inner peace [39]. Finally, a clear takeaway here is that survivors are "acting" even if we may not outwardly see those actions. By raising both survivor and practitioner awareness of these multifaceted healing efforts, we hypothesize there can be more meaningful therapeutic relationships by meeting survivors where they are in their recovery journey, understanding and acknowledging the strategies they are engaging in, while also brainstorming together about future help-seeking opportunities based on their healing goals.

The relationships among the recovery actions in this tool and barriers to help-seeking were not always negative. While our intuition would expect this relationship, these negative relationships were only seen between Building positive emotions and the barriers of Shame and Financial concerns. We also saw that higher PTSD scores were related to TRAC subscales. These findings may imply the central place of the trauma-related cognition of self-blame in shaping recovery action engagement behaviors. These findings are also consistent with the literature that shows that the attribution of the cause of the traumatic event to oneself may hinder women's engagement in recovery [71]. This suggestion aligns with previous studies which had demonstrated that women who experience GBV often fear that friends and family members would be unsupportive, judgmental, or critical when the abuse was disclosed [71–73] and that their blame induces indirect support-seeking such as seeking support without sharing the reason why [74].

Furthermore, the positive correlations between Establishing security with Feeling Frozen and Constraints also makes sense, because survivors often feel confused, stuck, and constrained by external forces, such as navigating a challenging legal system [75]. For example, many survivors anticipate adverse treatment by criminal/legal systems from the start, particularly those who hold identities that have been historically oppressed or abused by these systems [76–78]. Additionally, the attitudes and behaviors of agents of the criminal justice system are often perceived as destructive and discouraging to the victims, causing them more profound despair, and described as further victimization [79–81]. Thus, a possible interpretation is that our sample perceived that they could only trust themselves to achieve security and establish safety. This suggestion may imply the importance of women's sense of control over their recovery process [82]. Of course, these are hypotheses that would need to be tested in future studies. This pattern suggests that providers should find new ways to understand and evaluate help-seeking and recovery actions to understand these patterns in survivors navigating various sociocultural contexts.

Several limitations in our study should be noted. First, the recovery actions were measured using a dichotomous scale. Future studies should aim to measure more fully how these activities are helpful and satisfying by creating a Likert-style response anchored with "not necessary" to "satisfied with this strategy". Using a continuous scale can help researchers assess each factor's contribution to women's states in longitudinal studies. Second, the study's cross-sectional design did not examine the longitudinal analysis of the relationships between women's symptoms, perceived barriers, and recovery actions. Future research should evaluate what internal and contextual variables contribute to engagement in particular recovery strategies and how they have either helped or hindered individual recovery journeys. Third, our study was based on an American sample. Future studies should examine the factors and the relationships using large and representative samples among different cultures. Finally, a complete psychometric examination would require convergent validity, and we anticipate using other trauma-related coping inventories.

## 5. Conclusions

The current study focused on developing a new inventory to capture the women's multifaceted recovery actions after GBV and can serve as a potential tool to describe

women's recovery journey in future studies. This scale development responded to growing scientific studies highlighting the importance of active engagement in the GBV recovery process [26]. By expanding our understanding and measurement of help-seeking beyond disclosure and formal service use, we can better capture the immense amount of effort survivors often take to heal. We can also begin to understand and evaluate healing strategies outside of the formal and informal setting to understand how these play a role in the healing journey. This measure sought to give voice to their lived experience and the strategies they found most helpful by deriving this instrument from survivors' healing narratives. Measuring diverse recovery actions can move quantitative literature forward into a more survivor-centered direction, understanding and promoting help-seeking and long-term recovery in this population.

## 6. Patents

The Clinical Ethnographic Narrative Interview (CENI) is a licensed product. Licensing can be requested at: https://umich.flintbox.com/technologies/4dfd32ef-8a26-4532-80de-ed5e03912446 (accessed on 10 July 2021).

**Author Contributions:** Conceptualization, L.S., L.G. and D.M.S.A.; methodology, L.S., L.G. and D.M.S.A.; validation, L.S., L.G. and D.M.S.A.; formal analysis, L.S., L.G. and D.M.S.A.; writing—original draft preparation, L.S., L.G. and D.M.S.A.; writing—review and editing, L.S., L.G. and D.M.S.A.; visualization, L.S., L.G. and D.M.S.A.; supervision, D.M.S.A.; project administration, D.M.S.A.; funding acquisition, L.S. and D.M.S.A. All authors have read and agreed to the published version of the manuscript.

**Funding:** This research was funded by National Clinician Scholars Program and Institute for Research on Women and Gender.

**Institutional Review Board Statement:** The study was conducted according to the guidelines of the Declaration of Helsinki and approved by the Institutional Review Board of University of Michigan (HUM00144780, 9/18/2019; HUM00091662, 11/13/2014; and HUM00191183, 12/07/2020).

**Informed Consent Statement:** Informed consent was obtained from all subjects involved in the study.

**Data Availability Statement:** This data is available on request from the senior author.

**Conflicts of Interest:** The authors declare no conflict of interest.

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
