# Peer review of "The Trauma Recovery Actions Checklist: Applying Mixed Methods to a Holistic Gender-Based Violence Recovery Actions Measure"

_sexes, doi:10.3390/sexes2030029_

Round 1

Reviewer 1 Report

This study makes an important contribution by considering a wider variety of trauma recovery actions than previous studies and by placing all actions in a positive light. It begins with the “voices” of survivors, uses sound qualitative methods to organize those voices to create measurement items, and then tests the psychometrics and validity of the measure. Below are my comments, which focus on clarifications that might be useful and some additional publications to consider:

 “Theories and process models of recovery strategies [3, 4] have evolved from viewing women as passive victims to viewing them as actively engaging in a multitude of private and public strategies to prevent, manage, and escape violence, and 43 to recovery from trauma.” More recent and more comprehensive evidence of evolving views of victims as active exist, for example Anderson, D. K., & Saunders, D. G. (2003). Leaving an abusive partner: An empirical review of predictors, the process of leaving, and psychological well-being. Trauma, violence, & abuse4(2), 163-191.

P2: “Studies in traditional, collectivistic 57 societies documented a series of powerful cultural practices that inhibit women’s help-58 seeking and push them to adopt passive coping styles. These cultural practices include 59 degrading the women and their families’ reputations, demeaning the women’s social status, and social ostracism”. For the conclusion to be valid, the studies would need to compare collectivist and individualist societies.  It does not appear that such comparison was made.

“Those who seek help tend to seek family help.” Most reviews find that families and friends are most likely sought for help.

Excellent points are made about the problems with pathologizing trauma.

The sample of young women who experienced sexual assault may have qualitatively different recovery experiences than the other two samples. You might want to discuss somewhere the pros and cons of combining these diverse samples. Also, how well does the literature review apply to the sexual assault survivors, when much of the literature review seems to cover intimate partner violence? The psychological processes and help-seeking barriers of dating and acquaintance sexual assault victims may differ considerably from intimate partner violence victims.

The Clinical Ethnographic Narrative Interview does not seem to directly measure internal and external coping responses surrounding victimizations experiences.  Are these coping responses measured indirectly? If so, please explain.  If not, is this a limitation of the measure?

p. 5: Internal reliability is given for the quantitative measures. Giving evidence of validity would also be useful.

When you say you “carried out a predictive validity analysis of TRAC subscale scores with Symptom burden and Barriers to Help Seeking scales”, what specific predictions did you make? What rationale did you have for the predictions?

It might be helpful to clarify the meaning of some of the terms in the qualitative analysis, for example, “abstracting codes upwards”, “Reconciliation meetings were held within the team to synthesize and condense items to minimize participant burden . . .  “

p. 7: The title of Table 1 needs to state that factor loadings are being shown.

p. 8: Two sentences are repeated beginning with “The analysis found an advantage to the five-factor solution . . . “

In Table 2, I do not see “Cronbach’s alpha of the sub-scales and the complete 35-item scale” unless I do not recognize an acronym”

I do not see the correlations between the scales in Table 3. They are in Table 4. The presentation of predictive validity should not include the intercorrelations among the recovery actions.

Where do the following variables exist in Table 4: management beliefs, discrimination, or perceived unavailability?  Are they called something else?

p. 9: The following sentence needs clarification: “There were also weak positive significant correlations between Feelng Frozen, perceiving constraints, and Establishing Security.” Both feeling frozen and constraints are related to establishing security, but not to each other.

Why are three of the recovery actions positively correlated with PTSD?  Does a higher score on the PTSD scale mean more PTSD? Are the findings those that were expected?  Given the t-test results on PTSD, a higher PTSD score means recovery actions were taken.  These findings suggest that greater traumatization may lead to post-traumatic growth as indicated by the recovery actions.

10: You might consider tying the discussion of thriving to the concept of post-traumatic growth. Here is one review of the literature: Ulloa, E. C., Hammett, J. F., Guzman, M. L., & Hokoda, A. (2015). Psychological growth in relation to intimate partner violence: A review. Aggression and Violent Behavior25, 88-94.

p. 11: Normally, the limitations are described in the Discussion section and not in the Conclusion.

Author Response

Response to reviewer 1:

 “Theories and process models of recovery strategies [3, 4] have evolved from viewing women as passive victims to viewing them as actively engaging in a multitude of private and public strategies to prevent, manage, and escape violence, and 43 to recovery from trauma.” More recent and more comprehensive evidence of evolving views of victims as active exist, for example Anderson, D. K., & Saunders, D. G. (2003). Leaving an abusive partner: An empirical review of predictors, the process of leaving, and psychological well-being. Trauma, violence, & abuse, 4(2), 163-191.

Reply: I reviewed your suggested reference confirmed this is consistent.

P2: “Studies in traditional, collectivistic 57 societies documented a series of powerful cultural practices that inhibit women’s help-58 seeking and push them to adopt passive coping styles. These cultural practices include 59 degrading the women and their families’ reputations, demeaning the women’s social status, and social ostracism”. For the conclusion to be valid, the studies would need to compare collectivist and individualist societies.  It does not appear that such comparison was made.

Reply:  I revised this section to avoid a comparison between collective and individualistic societies.

“Those who seek help tend to seek family help.” Most reviews find that families and friends are most likely sought for help.

Reply: Clarified this in the text

Excellent points are made about the problems with pathologizing trauma. Thank you!

The sample of young women who experienced sexual assault may have qualitatively different recovery experiences than the other two samples. You might want to discuss somewhere the pros and cons of combining these diverse samples. Also, how well does the literature review apply to the sexual assault survivors, when much of the literature review seems to cover intimate partner violence? The psychological processes and help-seeking barriers of dating and acquaintance sexual assault victims may differ considerably from intimate partner violence victims.

Reply: I evaluated the references and confirmed that they were about sexual assault and IPV inclusively.

The Clinical Ethnographic Narrative Interview does not seem to directly measure internal and external coping responses surrounding victimizations experiences.  Are these coping responses measured indirectly? If so, please explain.  If not, is this a limitation of the measure?

Reply: I clarified the description of this instrument to reflect that it is about help seeking and recovery actions.

5: Internal reliability is given for the quantitative measures. Giving evidence of validity would also be useful. I clarified that language here.

When you say you “carried out a predictive validity analysis of TRAC subscale scores with Symptom burden and Barriers to Help Seeking scales”, what specific predictions did you make? What rationale did you have for the predictions?

Reply: The predictions are now clearly stated.

It might be helpful to clarify the meaning of some of the terms in the qualitative analysis, for example, “abstracting codes upwards”, “Reconciliation meetings were held within the team to synthesize and condense items to minimize participant burden . . .  “

Reply: I clarified our analysis method.

7: The title of Table 1 needs to state that factor loadings are being shown.

Reply: Revised.

8: Two sentences are repeated beginning with “The analysis found an advantage to the five-factor solution . . . “

Reply: Removed the redundant sentence.

In Table 2, I do not see “Cronbach’s alpha of the sub-scales and the complete 35-item scale” unless I do not recognize an acronym”

Reply: This is revised.

I do not see the correlations between the scales in Table 3. They are in Table 4. The presentation of predictive validity should not include the intercorrelations among the recovery actions.

Reply: This is corrected.

Where do the following variables exist in Table 4: management beliefs, discrimination, or perceived unavailability?  Are they called something else?

9: The following sentence needs clarification: “There were also weak positive significant correlations between Feeling Frozen, perceiving constraints, and Establishing Security.” Both feeling frozen and constraints are related to establishing security, but not to each other.

Reply:  These sentences are clarified.

Why are three of the recovery actions positively correlated with PTSD?  Does a higher score on the PTSD scale mean more PTSD? Are the findings those that were expected?  Given the t-test results on PTSD, a higher PTSD score means recovery actions were taken.  These findings suggest that greater traumatization may lead to post-traumatic growth as indicated by the recovery actions. We interpret this to mean that those with higher PTSD were searching for recovery actions that were effective and that might bring them peace. 

Reply: This discussion is clarified.

10: You might consider tying the discussion of thriving to the concept of post-traumatic growth. Here is one review of the literature: Ulloa, E. C., Hammett, J. F., Guzman, M. L., & Hokoda, A. (2015). This review is added. Psychological growth in relation to intimate partner violence: A review. Aggression and Violent Behavior, 25, 88-94.

Reply: This linkage is clarified.

11: Normally, the limitations are described in the Discussion section and not in the Conclusion.

Reply: This is revised accordingly.

Reviewer 2 Report

Review of manuscript: The Trauma Recovery Actions Checklist: Using Mixed Methods to a holistic Gender-Based Violence recovery actions measure

Comments for Authors: 

This is an interesting piece of research into gender-based violence recovery actions, but the paper has some major problems that need to be addressed.

Firstly, how the samples were selected in the qualitative phase? There have been an explanation of these samples, but no criteria for this selection were reported. This is important to decide whether the research results are reliable and credible.

Secondly, what are the statistic results of the online survey? Could the authors provide a table to show the details? This is crucial for readers to understand the following correlation and evaluation analysis of recovery actions.

Thirdly, the qualitative phase contributed to six central themes of lifestyle self-care, emotional self-care, relating to others, finding safety, finding peace, and futuring, why and how these themes were identified? And why did the trauma recovery action consist of five aspects? More information should be given.

Author Response

Firstly, how the samples were selected in the qualitative phase? There have been an explanation of these samples, but no criteria for this selection were reported. This is important to decide whether the research results are reliable and credible.

Reply: I clarified that this was the English language data from a larger international study.

Secondly, what are the statistic results of the online survey? Could the authors provide a table to show the details? This is crucial for readers to understand the following correlation and evaluation analysis of recovery actions.

Reply: The sample characteristics section was added including a sample table.

Thirdly, the qualitative phase contributed to six central themes of lifestyle self-care, emotional self-care, relating to others, finding safety, finding peace, and futuring, why and how these themes were identified? And why did the trauma recovery action consist of five aspects? More information should be given.

Reply: I reworked the qualitative analysis section to clarify our steps, and added detail in the exploratory factory analysis section to describe how the data reduction from our 6 qualitative to our final 5 factors was achieved.